# Synthesis and Biological Activities of Some Metal Complexes of Peptides: A Review

**DOI:** 10.3390/biotech13020009

**Published:** 2024-04-08

**Authors:** Petja Marinova, Kristina Tamahkyarova

**Affiliations:** Department of General and Inorganic Chemistry with Methodology of Chemistry Education, Faculty of Chemistry, University of Plovdiv, “Tzar Assen” Str. 24, 4000 Plovdiv, Bulgaria; chtamah@gmail.com

**Keywords:** peptides, metal complexes, biological activities

## Abstract

Peptides, both natural and synthetic, are well suited for a wide range of purposes and offer versatile applications in different fields such as biocatalysts, injectable hydrogels, tumor treatment, and drug delivery. The research of the better part of the cited papers was conducted using various database platforms such as MetalPDB. The rising prominence of therapeutic peptides encompasses anticancer, antiviral, antimicrobial, and anti-neurodegenerative properties. The metals Na, K, Mg, Ca, Fe, Mn, Co, Cu, Zn, and Mo are ten of the twenty elements that are considered essential for life. Crucial for understanding the biological role of metals is the exploration of metal-bound proteins and peptides. Aside from essential metals, there are other non-essential metals that also interact biologically, exhibiting either therapeutic or toxic effects. Irregularities in metal binding contribute to diseases like Alzheimer’s, neurodegenerative disorders, Wilson’s, and Menkes disease. Certain metal complexes have potential applications as radiopharmaceuticals. The examination of these complexes was achieved by preforming UV–Vis, IR, EPR, NMR spectroscopy, and X-ray analysis. This summary, although unable to cover all of the studies in the field, offers a review of the ongoing experimentation and is a basis for new ideas, as well as strategies to explore and gain knowledge from the extensive realm of peptide-chelated metals and biotechnologies.

## 1. Introduction

Nature has fine-tuned protein sequences, structures, and binding sites over thousands of years. Amino acid building blocks exhibit both chemical and structural versatility, efficiently and specifically binding to other biological macromolecules. Numerous peptides are either naturally occurring or derived from parent proteins through cleavage. As a result, peptides—whether natural, derived or synthetic—are exceptionally well-suited for various purposes and find applications across a wide range of scenarios [1] (see Figure 1).

Among the long list of applications of therapeutic peptides [2] are combating cancer [3], microbial infections [4] (Figure 2), viral infections [5], and neurodegenerative diseases [6], among others. Peptides can autonomously organize into various structures, such as nanostructures [7,8,9], microstructures [10], hydrogels [11], treatment of cancerous growths [12], and drug delivery [13].

Metal ions have the ability to facilitate the self-assembly of peptides, leading to the formation of supra-molecular structures. These structures are able to serve various functions, such as catalysts [14,15] or forming assemblies with nanocavities that can be modified, potentially enabling enantioselective recognition [16]. Figure 3, which is shown below, depicts how metal complexes and peptides create metal–peptide assemblies.

Peptides serve as recognition elements in molecular probes because of their specific binding capabilities [17,18]. Additionally, their capacity to bind to metals is employed for direct detection of those metals [19,20]. The structure of the gold complex is given in Figure 4 and the Cu(II) and Ni(II) complex in Figure 5, respectively. Metallopeptides are also utilized as electrical probes to identify bacteria [21,22].

Metal–peptide complexes offer chemical stabilization to the metal through chelation. Within the food industry, peptides function as chelators, enhancing the bioavailability of mineral supplements by safeguarding them against oxidation and altering their solubility properties [23,24].

Additionally, peptides’ metal-binding capacity has been harnessed to enhance the biocompatibility of gold nanoclusters for imaging [25], and bound lanthanide serves as an imaging agent [26]. Moreover, the specific binding between natural peptides and heavy metals can be utilized for metal remediation [27,28]. The structure and biosynthetic pathways of the complex are given in Figure 6.

The methods for synthesis, heavy metal toxicity, the biosynthesis of metallothioneins and phytochelatins in microalgae, the interaction of metal complexes with peptides, the functions and nature of zinc signals, structure of metal–peptides, relationship of SAR, and biological activity are reported in the following literature reviews: [1,24,25,27,29,30,31,32,33,34,35,36,37,38,39]. Some metal complexes can be used as potential therapeutics to treat Alzheimer’s [29,40,41,42], neurodegenerative disorders, Wilson’s [43], Parkinson’s disease [44,45] and Menkes disease [46]. The mechanisms of action of these compounds are different and include the modification of DNA/RNA, permeabilization, protein and cell wall synthesis, and modulation of gradients of cellular membranes. Table 1, shown below, provides a list of neurodegenerative disorders and a list of references to articles with a greater focus on them.

We hope that this review will help researchers whose main focus is on inorganic synthesis and reveal the hidden potential in ligand molecules like α-amino acids and peptides. It is worth mentioning that although this review covers literature published in the last two decades, a single review cannot cover all the obtained data. While choosing papers, we took into consideration the potential opportunities for biological activities that might be applicable to researchers in the field, as well as the nature of the transition metal. In the review, we discuss complexes of both traditionally popular metals and essential elements (Cu, Zn, Co, Fe) and non-essential elements (Pd, Pt, Ni, Ag and lanthanides).

It is for this reason that the data presented in this review are classified by the essential and non-essential elements in these ligands, and not by the nature of the ligand molecule.

### 1.1. Metal Complexes of Peptides with Some Essential and Non-Essential Elements

#### 1.1.1. Some Essential Elements

##### Copper (I)

Copper holds a crucial status as a metal that is essential for various biological functions, contributing significantly to both structural integrity and biochemical processes. Facilitating its transportation are specialized copper chaperones, a necessity due to their inherent toxicity, particularly stemming from their capability to transition between their reduced form, Cu(I), and oxidized state, Cu(II) [47]. In the realm of structural NMR (Nuclear Magnetic Resonance), the distinction between these two states is stark, as Cu(I) exhibits diamagnetic properties while Cu(II) manifests paramagnetic behavior. The handling of Cu(I) samples poses a primary challenge, demanding meticulous preparation within a glove box environment and ensuring airtight sealing to prevent oxidation upon exposure to air [48]. The structure of the copper complex is given in Figure 7.

In exploring the binding sites, both linear and cyclic peptides have served as valuable models, mimicking the conserved sequence essential for copper chaperone interactions. Investigation into their binding modes has been conducted through mutation studies and manipulation of environmental factors. For instance, altering the conserved Met residue within the binding sequence severely hindered copper complexation [44], while adjustments in pH levels led to varying binding modes [49]. These findings imply that pH conditions can influence the coordination environment of the metal, highlighting the utility of peptide–metal models in elucidating potential release mechanisms. The possible mechanism of action of copper complex [44] is given in Figure 8.

Further structural examinations focused on the interaction between Cu(I) and the N-terminal A 16 fragment of amyloid beta (A) and sought to ascertain the involvement of all three His residues in metal ion binding [50]. Notably, these studies were conducted on small peptides derived from the binding sites of larger molecules, underscoring their relevance in dissecting intricate molecular interactions.

##### Copper (II)

Copper(II) stands out as the heaviest among the paramagnetic essential metals, attracting considerable attention in studies involving various metal-binding proteins [51]. Research endeavors have employed signal broadening techniques to pinpoint the residues involved in binding [52], with such phenomena often offering insights into the sequence of binding events [53]. When proton (^1^H) spectra exhibit broadening effects, complementary information can be extracted from the corresponding carbon-13 (^13^C) spectra [54]. Typically, these investigations necessitate the initial assignment of the 1H spectrum followed by titration experiments with Cu(II) to identify the specific binding groups. Subsequently, the formation of the bound complex can be confirmed either through experimental means [53] or through computational simulations [55]. This allows for the positioning of the Cu(II) ion within the identified binding site, ultimately yielding a detailed depiction of the bound structure. Figure 9 show metal-binding proteins [51], while Figure 10 provides a schematic representation of the proposed structures [52].

Recently, Georgieva et al. proposed the formation of complexes between Cu(II) and hemorphin peptide derivatives in an aqueous solution with a pH of 10.98 [56]. Through meticulous examination of spectroscopic data and voltametric calculations, complete agreement has been reached, confirming the formation of stable copper(II) complexes with peptides in aqueous solutions. These complexes exhibit a 1:2 stoichiometry for Cu-H-V, Cu-AH-V, Cu-NH7C, and Cu-NCH7, while a 1:1 stoichiometry is observed for Cu-AC-V and Cu-C-V. Notably, the virucidal activity against human respiratory syncytial virus and adenovirus (HRSV-2) at 30 and 60 min remains weakly affected by these complexes, similar to the starting peptide compounds [56]. Figure 11 illustrates the structures of two of these newly identified complexes.

Hickey et al. documented the discovery of a copper complex featuring a styrylpyridine group, which exhibited an affinity for Aβ plaques within human brain tissue [42]. Figure 12 depicts the structure of these ^64^Cu complexes.

Given the challenges in directly measuring copper(II)-bound complexes, diamagnetic metals with square planar binding geometry often serve as substitutes for deriving the bound structure. Through line-broadening analysis, the position of copper binding can be inferred. If there is experimental evidence indicating similarity in binding modes, Cu(II) can be introduced into structure calculations, enabling the determination of copper(II)-bound structures. This methodology has been successfully applied to peptide–metal complexes utilizing palladium(II) [57], silver(I) [58], and nickel(II) [59,60].

Line-broadening analysis of proton (^1^H) spectra has been instrumental in identifying residues involved in binding Cu(II) to model copper(II)-binding sites [61], as well as in copper(II)-mediated aggregation processes [62]. Additionally, line-broadening effects in carbon-13 (^13^C) spectra can be utilized [63], while chemical shift deviations can be monitored through various experiments including ^1^H-^1^D and 2D ^15^N-HSQC, ^13^C-HSQC, and ^1^H-^1^H TOCSY experiments. Furthermore, precise determination of the binding nitrogen in histidine residues has been achieved through proton spin–lattice relaxation rate studies [64]. Sun et al. describes the synthesis, design and investigation of transition metal complexes with flexible histidine-containing peptides [65]. Copper(II) complexes of glycylglycylglycine peptides were also obtained [66]. Copper has the ability to amplify the antimicrobial activity of antibiotics, including viomycin and capreomycin. The complex of copper and viomycin has hydrogen bonds that make it stable and provide helical conformation in the peptide, making the DNA susceptible to strong degradation [67]; the described mechanism is connected to the modulation of the antigenomic δ-ribozyme catalytic activity [68].

##### Zinc (II)

Zinc stands out as a crucial ion involved in structural, catalytic, and regulatory functions within proteins, being present in approximately 10% of all known proteins [31,58]. Notably, zinc finger proteins represent a prominent category among these, showcasing their flexibility and stability through metal-binding interactions. To unravel the intricacies of zinc finger binding, both short and longer peptides have been extensively utilized. These peptides serve as invaluable tools for modeling various aspects of zinc finger behavior, including metal coordination, folding dynamics, and actual binding processes [69,70,71,72]. Figure 13 gives a comparison between the α-peptide and oligourea backbones, as well as an illustration of the structures.

In the realm of peptide modeling, cyclic peptides featuring linear tails have emerged as particularly promising candidates. These structures exhibit notable conformational and thermodynamic stability when compared to their linear counterparts, effectively mimicking the zinc-ribbon fold characteristic of zinc fingers. Moreover, cyclic peptides demonstrate enhanced binding affinity towards Zn(II) ions, underscoring their utility in zinc finger modeling [73]. Through peptide-based investigations, insights into secondary structural elements common in zinc fingers have been gleaned, shedding light on their pivotal role in both folding and zinc binding processes. Notably, peptide engineering efforts have led to the development of modified peptides that retain tertiary folds and stability comparable to their natural zinc finger protein counterparts, even with substantial alterations to their native residues [74]. Figure 14 presents an NMR confirmed structure of ZnL_ZR_.

The binding of zinc ions can induce significant conformational changes in peptides [75], a phenomenon exemplified by its ability to trigger the oligomerization of amyloid beta, a 42-amino acid polypeptide associated with Alzheimer’s disease. To study such processes effectively, truncated peptides preserving the beta-sheet formation region while minimizing complete oligomerization have been employed. For instance, investigations into the familial Taiwanese mutation D7H region of amyloid beta, which impacts zinc-induced oligomerization, have been facilitated through the use of stable homodimers formed via zinc binding [40,41]. Figure 15 below represents the arrangement of Zn ions in solution within the rat Aβ(1–16) dimer complexes.

Furthermore, exploring the resistance of rats to Alzheimer’s disease has provided valuable insights into zinc-induced dimerization within amyloid beta proteins. Utilizing truncated peptides, researchers have elucidated the interface involved in zinc-induced dimerization, showcasing a potential avenue for the rational design of drug compounds aimed at disrupting plaque-forming processes.

##### Cobalt (II) and (III)

Cobalt, existing in both its (II) and (III) oxidation states, though present in trace amounts within the body, holds vital significance as an essential element. However, when present in excess, cobalt poses toxicity risks attributed to its capacity for generating reactive oxygen species and displacing iron in metalloenzymes, thereby rendering them inactive [33]. Predominantly paramagnetic, cobalt manifests distinct characteristics in its (III) state, where it can adopt both high- and low-spin configurations, the latter being diamagnetic [33]. For instance, the elucidation of a diamagnetic Co(III) complex with a peptide-porphyrin conjugate was achieved through conventional methodologies [76]. Notably, the large paramagnetic chemical shifts (PCSs) and minimal paramagnetic relaxation enhancements (PRE) exhibited by high-spin Co(II) render it conducive to structure determination [32]. Figure 16, Figure 17, and Figure 18 depict the structure of Co(III) complexes including light-activated Co(III) prodrugs; the general mechanism of action of cobalt complexes for delivering drugs through reduction activation; the molecular structure of Co(III)-mimochrome IV in stereo view, representing the average structure derived from both NMR experimental data and RMD calculations, respectively.

Similarly, the binding site of a cobalt(III)–Schiff base complex to another amyloid beta protein fragment was elucidated using ^1^H-NMR, with binding histidines identified through line-broadening effects [77]. Figure 19 presents transition metal complex suggestions.

Further insights into cobalt binding were gleaned from an investigation into cobalt(II) binding to fibrinopeptide B, a factor implicated in thrombosis. Although the peptide structures were determined in the presence of cobalt(II) and gadolinium(III), the metal ions were not visualized within the structures [78]. Moreover, a 30-amino acid peptide initially recognized for its manganese-binding capabilities, derived from a decapeptide repeat in the calcium protein Cap43, was also found to exhibit affinity for Co(II). Line-broadening observed during the interaction enabled the identification of binding histidine residues, while the pH range conducive to binding was elucidated as part of the binding mechanism. Subsequent modeling efforts generated and minimized postulated coordination spheres of Co(II) based on NMR data, utilizing software like HyperChem™ [79]. Figure 20 displays the model depicting the probable coordination spheres of Co(II) with the multi-histidine peptide fragment.

Additionally, the conformational alterations of peptides upon binding to chiral cobalt oxide nanoparticles were investigated to explore chiral evolution phenomena. Tripeptide ligands exhibited distinct peak sets attributed to the high PCS of the nanoparticles, facilitating structure determination owing to their minimal PRE effects [80].

##### Iron (II) and (III)

Iron stands as an indispensable element crucial for oxygen transport within the body, with both excess and deficiency leading to various disorders [34,35]. Existing in both ferric and ferrous forms, iron exhibits paramagnetic properties, and the interconversion between these forms plays a central role in its functionality. Given the facile oxygenation of ferrous to ferric iron ions, meticulous sample preparation within an oxygen-free glove box environment is imperative to mitigate undesired reactions. The following studies underscore innovative strategies for investigating iron, highlighting the efficacy of a multidisciplinary approach. Figure 21 below illustrates the circulation and distribution of iron within the human body.

Detection of iron binding is notably achieved through the observation of line-broadening effects proximal to the binding site. For instance, a study involving the grafting of a six-residue iron-binding motif onto a 29-residue peptide utilized NMR spectroscopy to discern line-broadening phenomena indicative of specific interactions between the peptide and Fe(III) ions. Complementary techniques such as circular dichroism, isothermal titration calorimetry, capillary zone electrophoresis, thermal denaturation, and computational modeling were employed to elucidate the binding mode and structure of the peptide model system [81]. In magnetotactic bacteria, biomineralization processes are governed by magnetite-associated proteins featuring short sequences capable of binding iron. Peptides derived from these iron-binding regions were subjected to reactions with Fe(II), Fe(III), Ni(II), and Zn(II) ions, enabling the determination of specificity, binding coefficients, and key binding residues through NMR spectroscopy. Subsequent coprecipitation assays were employed to ascertain the significance of each binding residue [82]. Additionally, computational methodologies were integrated with NMR spectroscopy to elucidate the structures of artificial peptides forming chiral helicate complexes with Fe(II) and Co(II) ions [83]. Furthermore, the utility of 1D and 2D NMR techniques capable of directly detecting paramagnetic complexes was demonstrated using the eight-amino acid microperoxidase-8 bound with heme iron as a model peptide for the cytochrome C binding site. Excitation sculpting with gradients was employed to suppress water signals in Fe(II)-bound samples for acquiring 1D spectra, while a superWEFT pulse sequence was utilized for measuring Fe(III)-bound samples [84,85].

#### 1.1.2. Some Non-Essential Elements

A variety of non-essential metals have been implicated in contributing to the pathogenesis of Alzheimer’s disease, prompting investigations into their interactions with amyloid beta-derived peptides to unravel potential mechanisms of action. For instance, studies have explored the interactions of aluminum and palladium with these peptides [57,86]. Figure 22 depicts the overlap of Aβ12 structures with and without, and Figure 23 represents the three-dimensional arrangement of the Pd(Aβ4-16) complex.

Nickel NMR measurements serve as a valuable tool for detecting the binding of nickel ions in diamagnetic form, thereby serving as a model for understanding the behavior of paramagnetic ions that may otherwise pose challenges for structural characterization [87]. Studies comparing different forms of nickel-binding peptides offer insights into the role of nickel complexation in peptide structure and hint at the potential roles of paramagnetic ions. The amino-terminal copper- and nickel-binding (ATCUN) motif, found in numerous proteins, has been extensively investigated using linear and cyclic peptide models with various divalent ions, including Co(II), Ni(II), and others [50,88]. Nickel itself has been implicated in metal-induced toxicity and carcinogenesis, prompting studies on Ni(II)-peptide models derived from the C-terminal of histone H2B to elucidate potential roles in carcinogenesis [89]. Additionally, interactions of Ni(II) with peptides derived from the human Toll-like receptor (hTLR4) have been studied due to nickel’s ability to elicit allergic responses [90]. Figure 24 illustrates the configuration of the Pd(II) complex [58], and Figure 25 shows the stereo view of the superimposed 20 lowest energy structures of Ni^2+^-H2B_105–112_ obtained from NMR data [89].

Gallium, a diamagnetic ion, has been shown to effectively substitute Fe(III) binding in Fe2S2 clusters while preserving the overall structure [91]. Palladium, also diamagnetic, has been utilized as a model for Cu(II) binding in amyloid beta-derived peptides and prion protein peptide derivatives due to its similar square planar geometry [57,92]. Investigations into the pH dependence of palladium coordination using Pd(II)-peptide complexes have further expanded our understanding [93]. Tsiveriotis et al. presented the interaction of histidyl containing peptides with Pt(II) and Pd(II) [94]. Figure 26 depicts the chelate structure of metal complexes containing Pt(II) and Pd(II), and Figure 27 represents the configuration of certain metal complexes containing Pd(II).

New square planar bis-chelated palladium amino acid complexes with proline and proline homologs were obtained [95]. The structures of novel catalyst complexes were established by X-ray analysis. Figure 28 presents the chelate structure of metal complexes containing Pd(II).

Silver, a toxic diamagnetic ion, interacts with the human copper transporter 1 (hCtr1). Structural studies involving an Ag(I) complex with a micelle-bound peptide derived from hCtr1 have suggested that the membrane surface may influence the structure of the extracellular domain of the protein and its binding to Ag(I) [96]. Similarly, studies on peptides derived from the human copper transporter 2 (hCtr2) have shown that Ag(I) binding occurs when the peptide is in its trimeric form [58]. Figure 29 displays a collection of backbone atoms from 20 structures with the most favorable target functions for the wild-type peptide in SDS micellar solution.

Platinum, another diamagnetic metal with potent anticancer properties, has been studied for its binding with a peptide comprising the receptor binding sequence of transferrin, employing titration followed by NMR to determine the binding [97].

Lanthanides, mostly paramagnetic, are frequently used as shift reagents to monitor significant changes in chemical shift that occur in their proximity. Studies involving lanthanides and peptides have aimed at conjugating lanthanide–peptide complexes to proteins to leverage the shift reagent properties on the protein structure. For example, NMR-derived structures of fibrinopeptide B in the presence of salts, including Ga(III), have been elucidated [89]. Additionally, investigations into the binding properties of small peptides derived from Ca(II)-binding sites with toxic lanthanides such as La(III), Eu(III), and Tb(III) have been conducted to evaluate their potential for diagnostic use as contrast agents [98]. The structure of La(III) complexes [98] is given in Figure 30. The metal complexes of Y(III), La(III), Ce(III), Pr(III) and Nd(III) with glycyi-L-proline were synthesized by Sandhu et al. [99]. The summary table with some essential and non-essential elements is given in Table 2.

A series of solid research works is devoted to the synthesis and characterization of transition metal complexes with amino acids and peptides. Commonly, such complexes are synthesized using the approaches of classical chemistry of non-organometallic compounds.

The summary data on the structure of the complexes and the donor atoms involved in the coordination are given in Table 3.

## 2. Conclusions

In summary, essential metals play pivotal roles in various biological processes, ranging from oxygen transport to enzymatic catalysis. Their involvement in structural integrity, regulation, and signaling underscores their indispensability for life. While their deficiency can lead to debilitating disorders, excess levels can result in toxicity and disease. Through meticulous research spanning diverse methodologies, scientists have elucidated the intricate mechanisms underlying the interactions of essential metals with peptides and proteins, shedding light on their physiological functions and potential implications in diseases such as Alzheimer’s. Furthermore, ongoing exploration of metal-peptide interactions holds promise for the development of novel therapeutic strategies and diagnostic tools. Thus, understanding the roles of essential metals in biological systems remains a cornerstone in advancing our knowledge of human health and disease.

In conclusion, the intricate interplay between non-essential metals and peptides underscores the multifaceted nature of their interactions and their potential implications in various biological processes and diseases, such as Alzheimer’s disease and carcinogenesis. Through diverse methodologies ranging from NMR spectroscopy to computational modeling, researchers have elucidated the binding modes, structural changes, and functional consequences of these interactions. Investigations into metals like nickel, palladium, silver, and platinum have provided valuable insights into their roles in toxicity, carcinogenesis, and therapeutic potential. Furthermore, studies involving lanthanides have offered novel approaches for utilizing their paramagnetic properties as shift reagents for structural analysis. Continued exploration of these metal–peptide interactions promises to deepen our understanding of their biological significance and may pave the way for the development of novel diagnostic and therapeutic strategies in various fields of medicine and biotechnology.

## Figures and Tables

**Figure 1 biotech-13-00009-f001:**
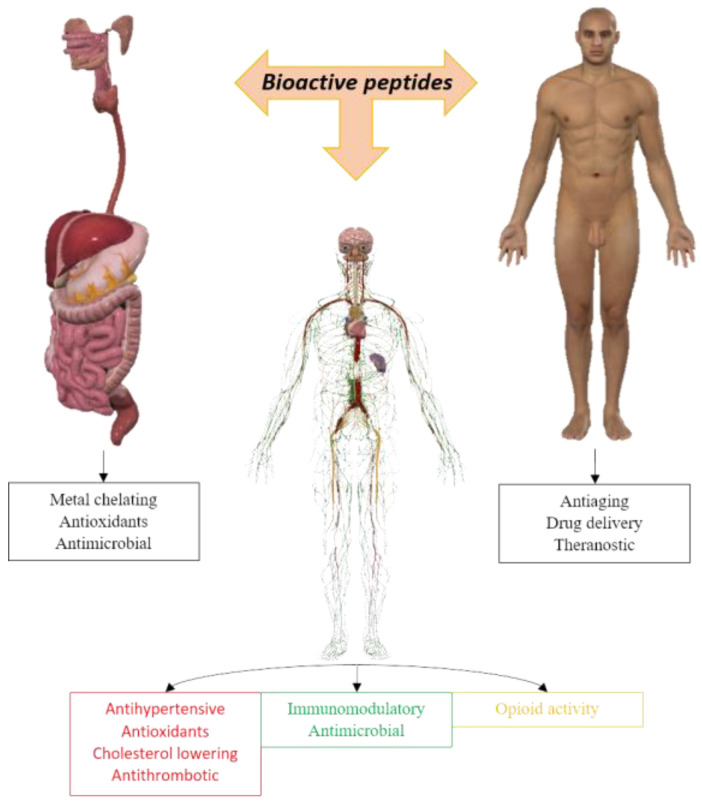
Different applications of bioactive peptides for humans.

**Figure 2 biotech-13-00009-f002:**
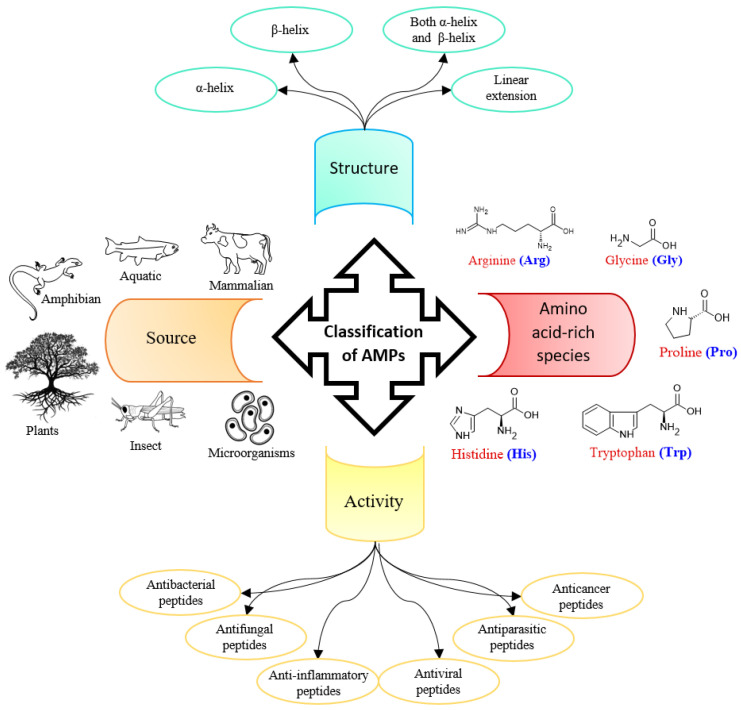
Classification of antimicrobial peptides.

**Figure 3 biotech-13-00009-f003:**
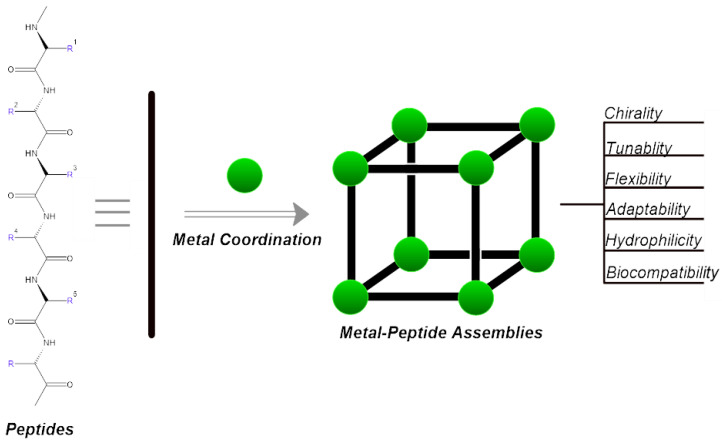
Metal complexes with peptides.

**Figure 4 biotech-13-00009-f004:**
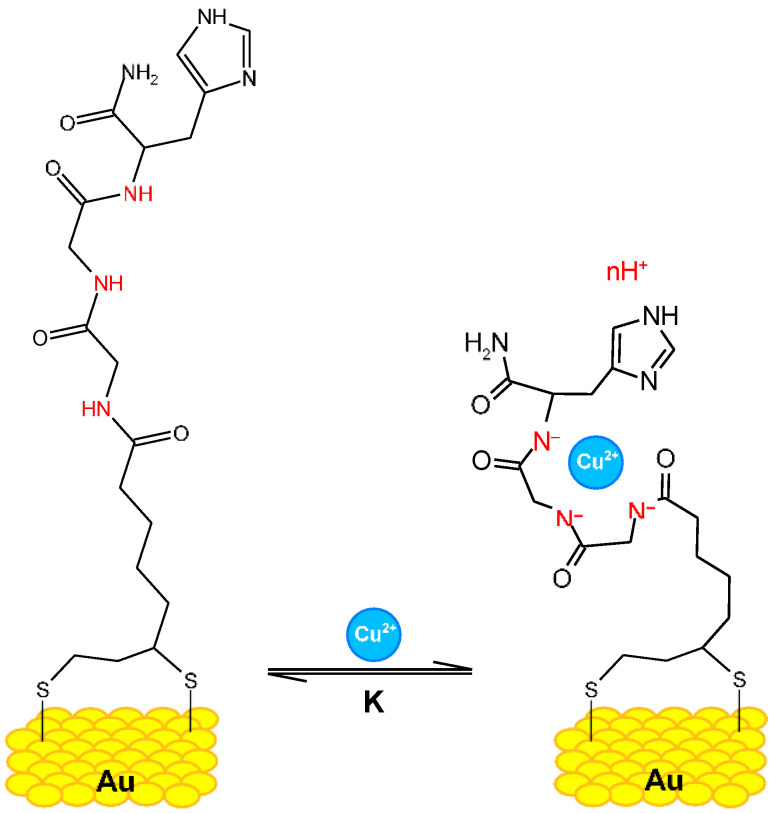
Glycine–glycine–histidine (Gly–Gly–His, GGH) monolayer on a gold surface with Cu^2+^ ions. Secondary amines, carrying different charges depending on the electrolyte’s pH, are indicated in red Reprinted/adapted with permission from Ref. [19]. 2019, Synhaivska et al. [19].

**Figure 5 biotech-13-00009-f005:**
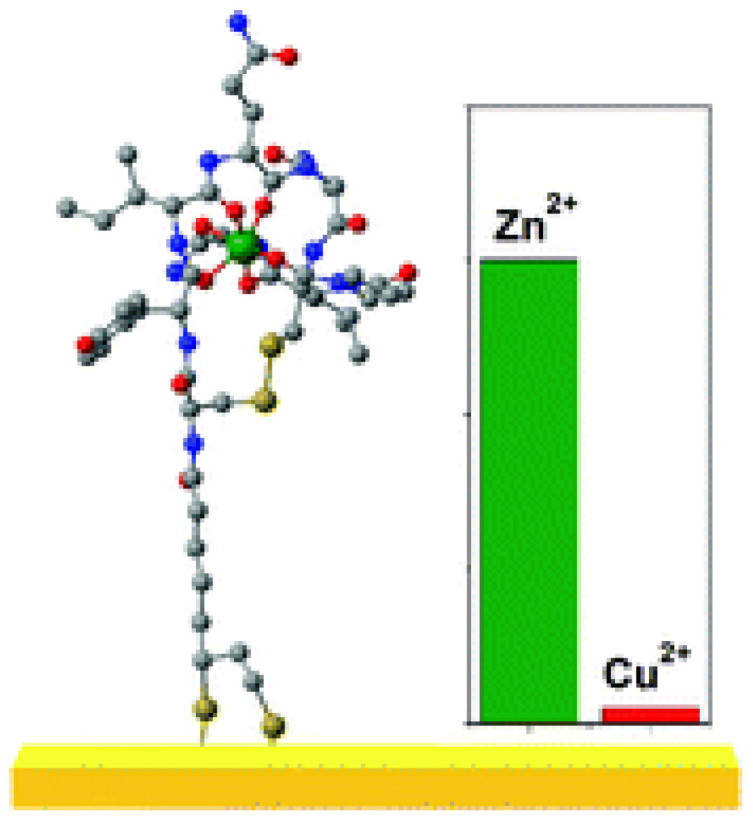
A new Zn^2+^ selective oxytocin-based biosensor that utilizes the terminal amine for surface anchoring, also preventing the response to Cu^2+^ Reprinted/adapted with permission from Ref. [20]. 2019, Mervinetsky et al. [20].

**Figure 6 biotech-13-00009-f006:**
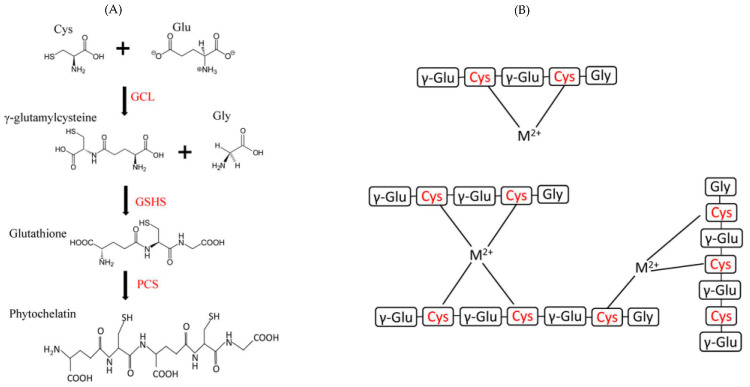
The structure and biosynthetic pathways of glutathione (GSH) and phytochelatin (PC) are explained, utilizing three-letter amino acid codes for abbreviations. (**A**) In GSH and PC biosynthesis, there are two and three reactions, respectively. Initially, a cysteine unit binds with the carboxylic group of the side chain of a glutamic acid residue to form γ-glutamylcysteine, catalyzed by glutamate–cysteine ligase (GCL). Then, γ-glutamylcysteine combines with a glycine residue to produce glutathione, catalyzed by glutathione synthetase (GSHS). PC-synthase (PCS) can potentially bind two or more GSH units to create phytochelatins. (**B**) Additionally, examples of PC–metal complexes involving divalent metal cations (M^2+^) are presented, with cysteine residues highlighted in red Reprinted/adapted with permission from Ref. [27]. 2020, Balzano et al. [27].

**Figure 7 biotech-13-00009-f007:**
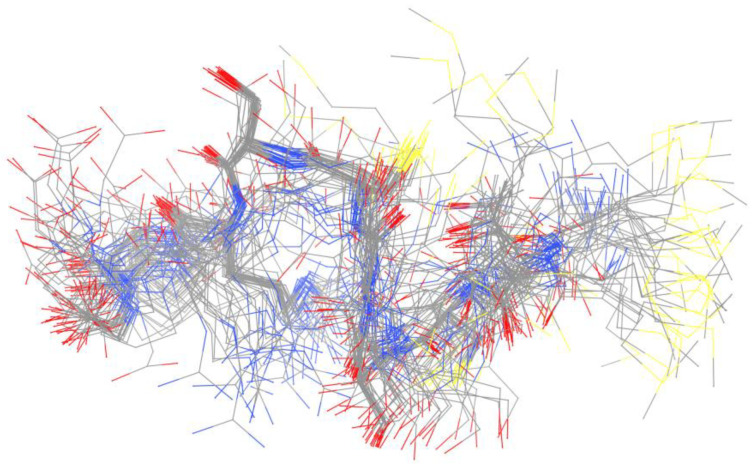
Ensemble derived from NMR data of copper-bound peptide. Complete 50-member ensemble representing all sample conformations superimposed on backbone Reprinted/adapted with permission from Ref. [48]. 2013, Shoshan et al. [48].

**Figure 8 biotech-13-00009-f008:**
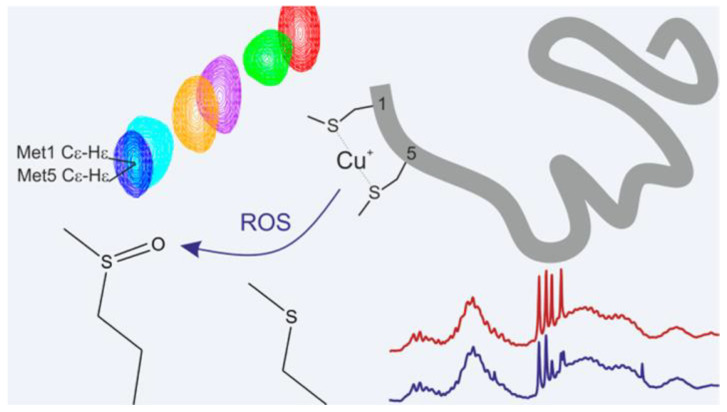
Possible mechanism of action of copper complex Reprinted/adapted with permission from Ref. [44]. 2014, Miotto et al. [44].

**Figure 9 biotech-13-00009-f009:**
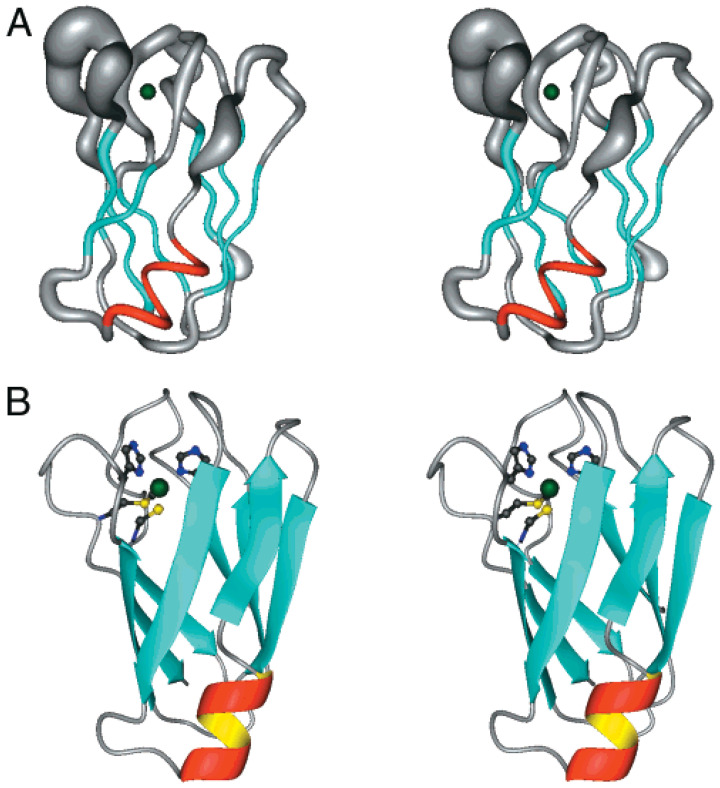
Metal-binding proteins presented by Bertini et al. MOLMOL^59^ stereo representations of (**A**) the “sausage” diagram of the superimposed 35 DYANA backbone structures of oxidized Synechocystis PCC6803 plastocyanin; (**B**) the restrained energy-minimized DYANA mean structure of oxidized Synechocystis PCC6803 plastocyanin, showing the elements of the secondary structure in different colors. The copper atom is shown in green at the top of the model. The four ligands (His39, Cys83, His86, and Met91) are represented as ball-and-stick colored according to the CPK code Reprinted/adapted with permission from Ref. [51]. 2011, Bertini et al. [51].

**Figure 10 biotech-13-00009-f010:**
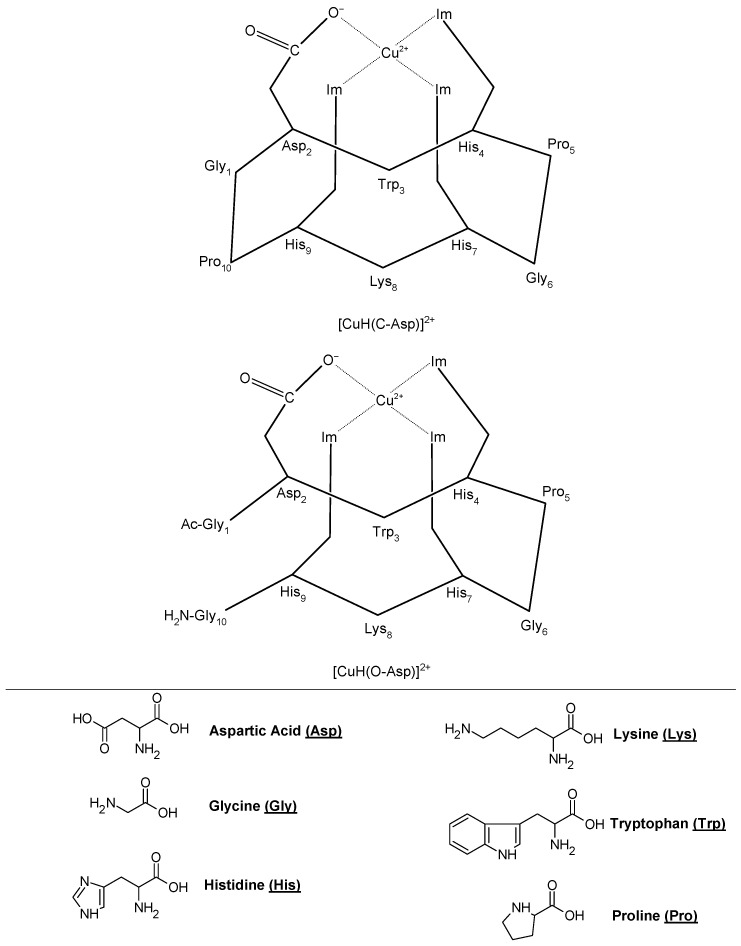
Schematic representation of the proposed structures (Im = imidazole) Reprinted/adapted with permission from Ref. [52]. 2015, Fragoso et al. [52].

**Figure 11 biotech-13-00009-f011:**
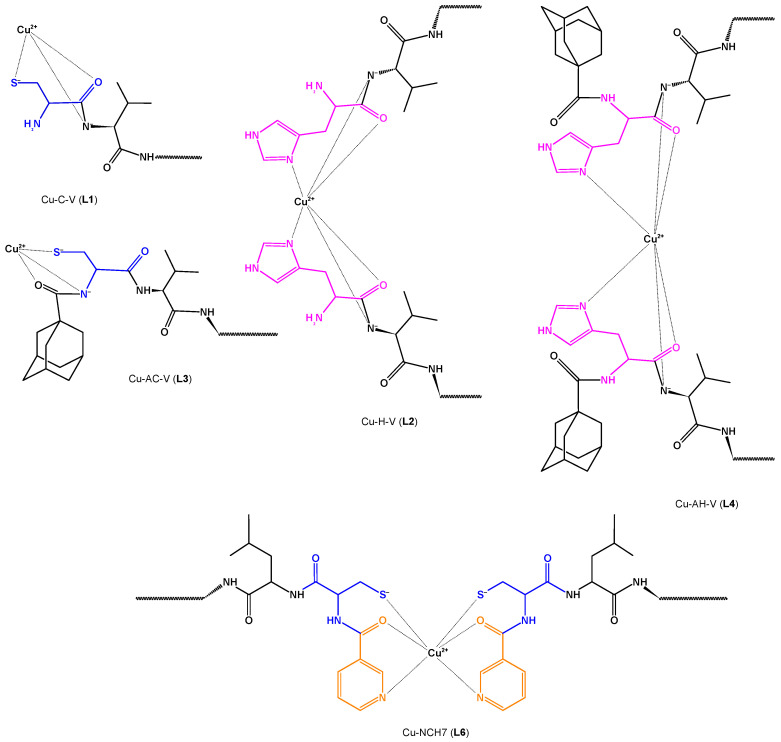
Schematic structure of Cu(II)-peptide complexes Cu-C-V(L1), Cu-H-V(L2), Cu-AC-V (L3), Cu-AH-V (L4), and Cu-NCH7 (L6) Reprinted/adapted with permission from Ref. [56]. 2023, Georgieva et al. [56].

**Figure 12 biotech-13-00009-f012:**
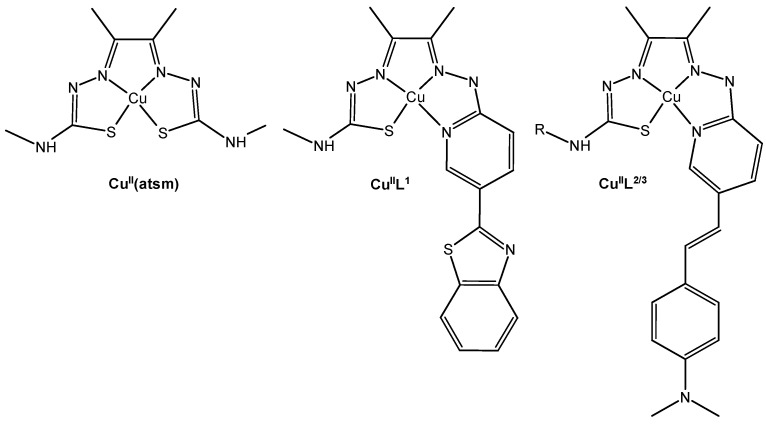
Structures of bis(thiosemicarbazonato)Cu^II^ complexes Cu^II^(atsm) and new hybrid thiosemicarbazonato-pyridylhydrazide Cu^II^ complexes with Aβ plaque targeting benzothiazole (CuIIL1) and styrylpyridine functional groups (Cu^II^L^2^ R = CH_3_; Cu^II^L^3^ R = CH_2_CH_2_N(CH_3_)_2_) Reprinted/adapted with permission from Ref. [42]. 2013, Hickey et al. [42].

**Figure 13 biotech-13-00009-f013:**
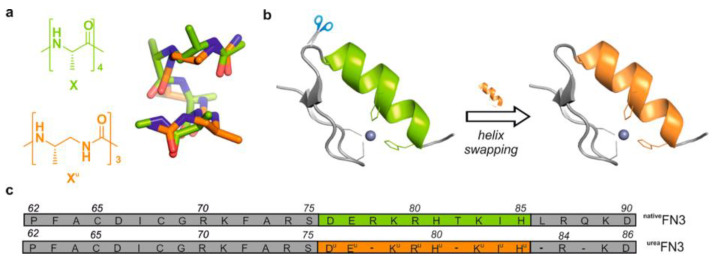
Design guided by the structure of composite proteins containing artificial helical foldamer region. (**a**) Comparison of α-peptide (green) and oligourea (orange) backbones and overlay of their helical structures (α-helical tetrapeptide vs. 2.5-helical triurea) illustrating the structural resemblance (dimensions, handedness, polarity) between the two helical backbones. (**b**) Application to the construction of a composite Cys2His2 zinc finger with a ββα fold. (**c**) Sequence of designed composite zinc finger ^urea^FN3 and the native zinc finger ^native^FN3 derived from Zif268 Reprinted/adapted with permission from Ref. [72]. 2019, Lombardo et al. [72].

**Figure 14 biotech-13-00009-f014:**
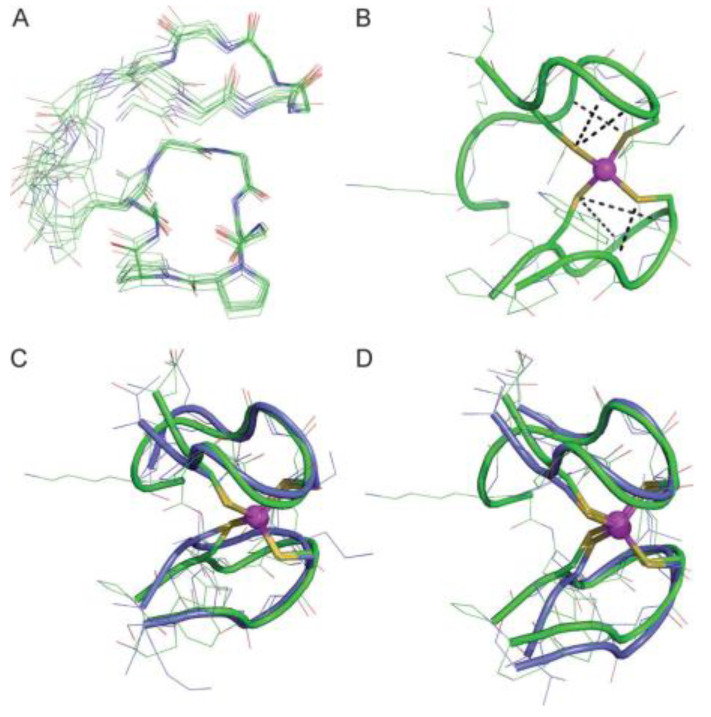
Solution structure of ZnL_ZR_ (pH 6.3, 298 K) deduced from NMR studies. (**A**) Superimposition of the 10 lowest energy structures calculated using XPLOR. All side chains except those of the ^D^Pro-Pro motif and that of the lysine serving as branching point between the cycle and the tail were removed for clarity. (**B**) Lowest-energy structure with the six NH···S hydrogen bonds displayed as black dashed lines. (**C**) Superimposition of the lowest-energy structure of Zn·L_ZR_ (green) with the zinc-ribbon domains of the subunit 9 of *Thermococcus celer* RNA polymerase II (blue, pdb 1QYP) and (**D**) *Clostridium pasteurianum* Zn loaded rubredoxin (blue, pdb 1IRN) Reprinted/adapted with permission from Ref. [73]. 2013, Jacques et al. [73].

**Figure 15 biotech-13-00009-f015:**
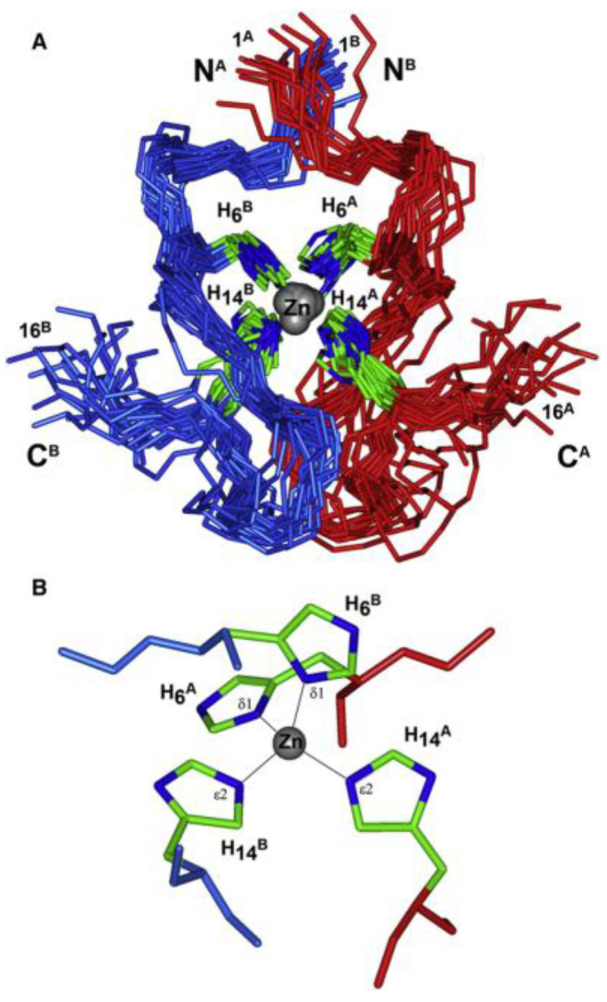
Structure of the rat Aβ(1–16) dimer complexes with Zn ions in solution (the 20 NMR conformers have been deposited in the Protein Data Bank with accession code 2LI9). Only the backbone atoms (Cα, C, and N) and the side chains of the His residues are shown. Chains A and B of the dimer are shown in red and blue, respectively. The N- and C-termini of both chains are labeled. (**A**) The family of 20 calculated NMR structures. (**B**) Coordination of the zinc ion by the histidine residues in the representative structure after additional QM/MM geometry optimization. The average distance between the Zn^2+^ ion and the nitrogen atoms of the His residues (Nd1 and Nε2) is 2.07 ± 0.05 Å Reprinted/adapted with permission from Ref. [40]. 2012, Istrate et al. [40].

**Figure 16 biotech-13-00009-f016:**
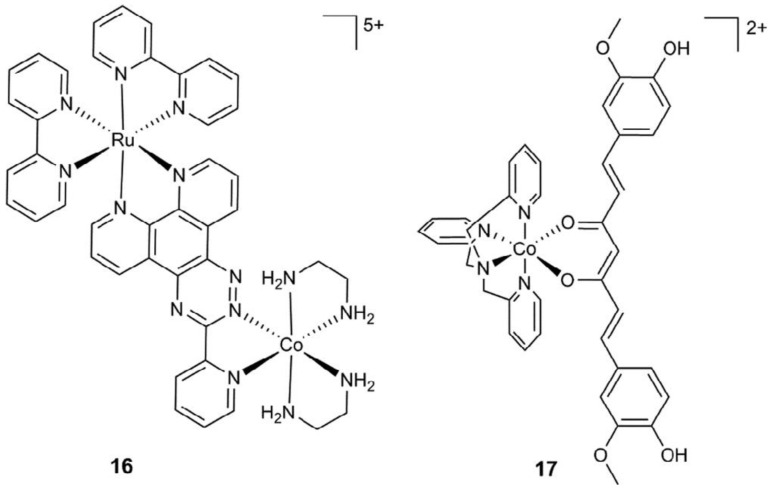
The structure of some Co(III) complexes including light-activated Co(III) prodrugs Reprinted/adapted with permission from Ref. [33]. 2018, Renfrew et al. [33].

**Figure 17 biotech-13-00009-f017:**
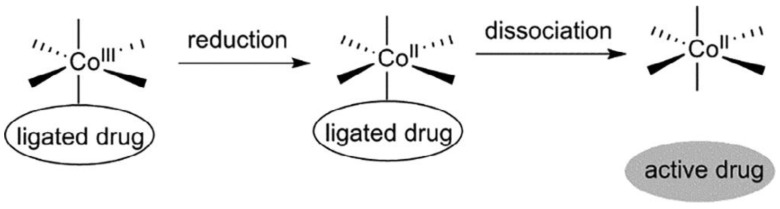
Generalised mechanism of action of cobalt complexes for reduction-activated drug delivery Reprinted/adapted with permission from Ref. [33]. 2018, Renfrew et al. [33].

**Figure 18 biotech-13-00009-f018:**
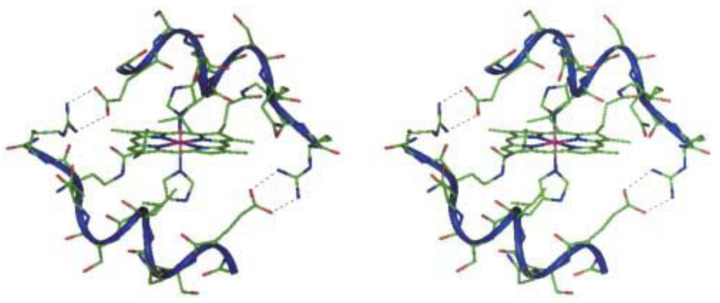
Molecular structure of Co(III)-mimochrome IV. Stereo view of the average structure, as obtained from NMR experimental data and RMD calculations Reprinted/adapted with permission from Ref. [76]. 2003, Lombardi et al. [76].

**Figure 19 biotech-13-00009-f019:**
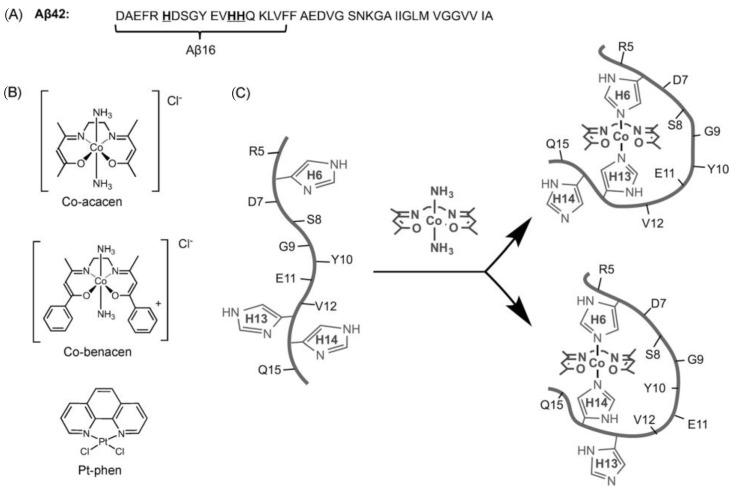
Transition metal complexes proposed by Heffern et al. (**A**) Amyloid-b (Ab) peptides used in these studies. Ab16 was used as a soluble peptide model to understand the interactions between Co–sb complexes and the N terminus of the Ab peptide. Ab42 was used to understand the effects of Co–sb complexes on peptide oligomerization and synaptic binding. (**B**) Transition metal complexes and naming schemes used in this work. Co–acacen and Co–benacen are Co–sb complexes that can coordinate His residues through dissociative ligand exchange of the axial ammines. The behavior of Co–sb on modulating Ab was compared to Pt–phen, a Pt^II^ complex previously shown to disrupt Ab-induced neurotoxicity. (**C**) Proposed scheme of the modulation of Ab activity by Co–acacen. Co–acacen is believed to coordinate the His residues of Ab through the two axial positions. Computational studies suggest the simultaneous coordination of His6 and either His13 or His14 as the most stable conformation. His coordination alters the Ab structure, disrupting oligomerization pathways and synaptic binding Reprinted/adapted with permission from Ref. [77]. 2014, Heffern et al. [77].

**Figure 20 biotech-13-00009-f020:**
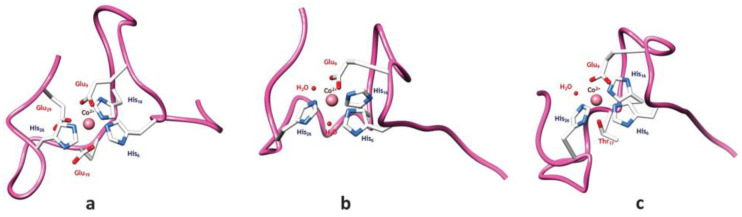
Model of the most likely coordination spheres of Co(II) with the multi-histidine peptide fragment: (**a**) Co-{3Nε(Im)-3O(Glu-COO^−^)}, (**b**) Co-{3Nε(Im)-3O(1Glu-COO^−^, 2H_2_O)}; (**c**) Co-{3Nε(Im)-3O(1 Glu-COO^−,^ 1 Thr-O, 1 H_2_O)} Reprinted/adapted with permission from Ref. [79]. 2013, Peana et al. [79].

**Figure 21 biotech-13-00009-f021:**
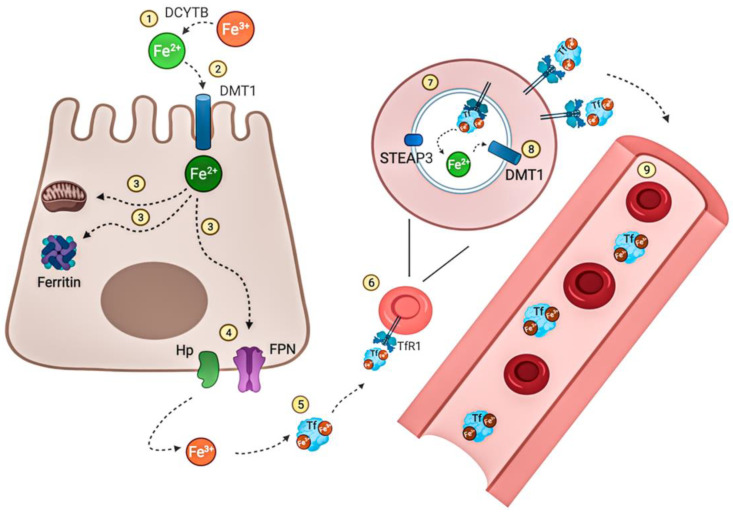
Iron distribution and circulation. Nonheme dietary iron Fe^3+^ is reduced to Fe^2+^ by the iron reducing DCYTB (1) prior to its uptake at the apical membrane of enterocytes via DMT1(2). Fe^2+^ can then be directly used for intracellular mechanisms, stored when bound to ferritin or released directly into the circulation (3). (4) Therefore, reduced iron Fe^2+^ is transported by ferroportin (FPN), the only known iron exporter so far, and then oxidized by hephaestin Hp to be then bound to Tf (5). Most of the iron resent in the circulation is bound to Tf. As a result, erythrocyte precursors (erythroblasts) take up this transferrin-bound iron via TfR1(6). Fe^3+^ bound to transferrin is reduced in the endosome by ferrireductase STEAP3 to Fe^2+^ (7) where it is exported via DMT1 (8) into the cytosol and enters the labile iron pool. Mature RBCs circulate in the blood for around 120 days (9) until they are removed from the circulation during rythrophagocytosis. The illustration was created using BioRender.com (accessed on 3 April 2021) Reprinted/adapted with permission from Ref. [35]. 2021, Vogt et al. [35].

**Figure 22 biotech-13-00009-f022:**
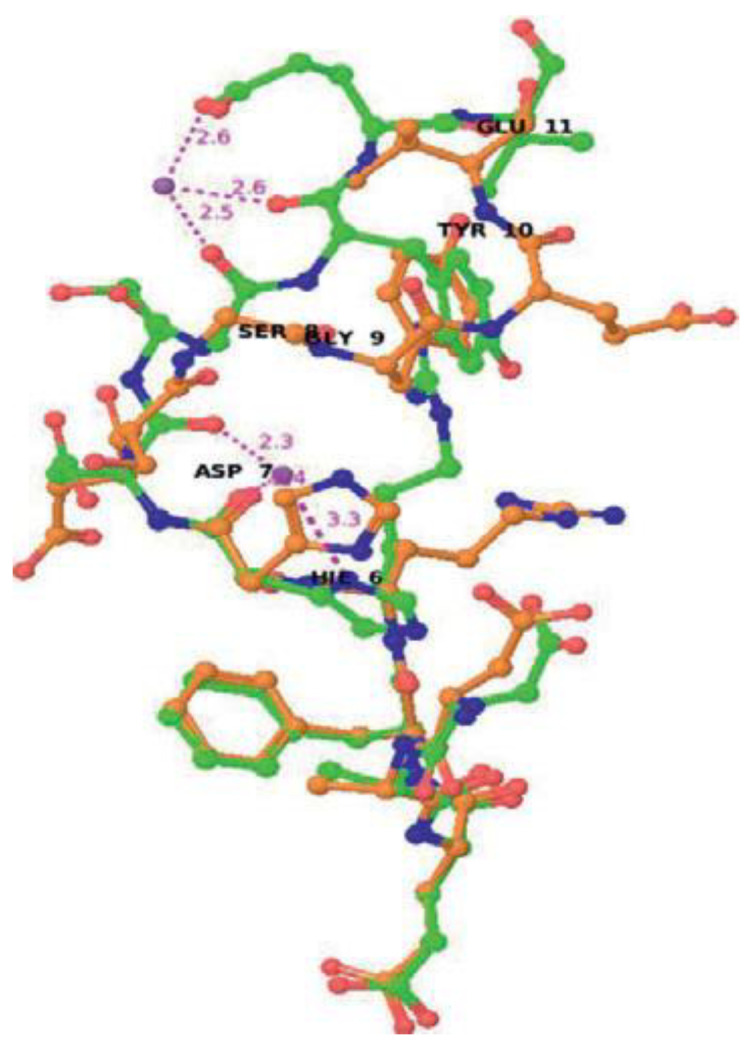
The structures of Ab12 with (green) and without metal (orange). Proposed binding sites of aluminium metal are also shown. C-terminus binding site has relatively lower energy, and is also compatible with NMR studies Reprinted/adapted with permission from Ref. [86]. 2013, Narayan et al. [86].

**Figure 23 biotech-13-00009-f023:**
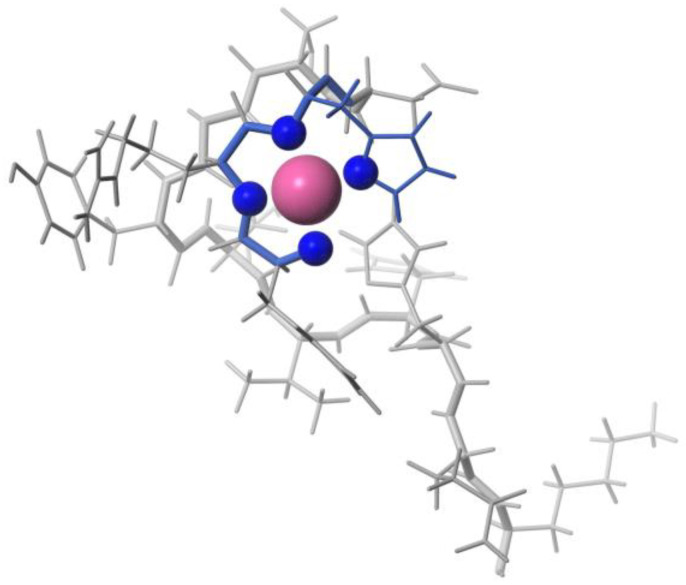
The 3D structure of Pd(Aβ4-16) complex included the N-terminal ATCUN/NTS motif binding the Pd(II) ion Reprinted/adapted with permission from Ref. [57]. 2020, Mital et al. [57].

**Figure 24 biotech-13-00009-f024:**
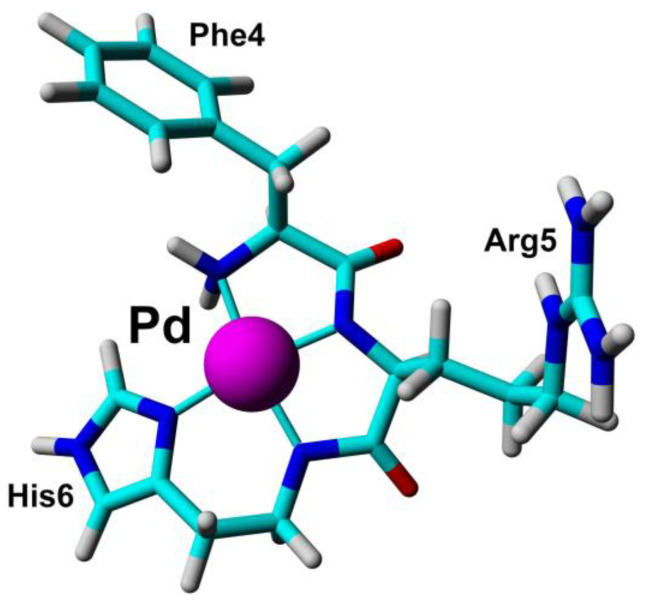
The structure of Pd(II) complex reported by Mital et al. The 3D structure of Phe-Arg-His-amide (FRH) peptide represented the N-terminal ATCUN/NTS motif saturated with Pd(II) ions based on collected NMR constraints and crystallographic data available for GGH tripeptides Reprinted/adapted with permission from Ref. [57]. 2020, Mital et al. [57].

**Figure 25 biotech-13-00009-f025:**
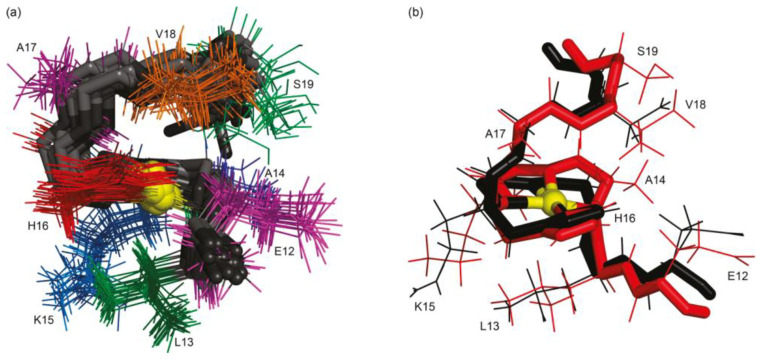
(**a**) Stereoview of superposition of the 20 lowest energy structures of Ni^2+^-H2B_105–112_ obtained from NMR data; (**b**) overlaid mean (black, E = 9460.38 kcal mol^−1^/gradient = 957.7 kcal mol^−1^ Å ^−1^) and geometric optimized (red, E = 168.6 kcal mol^−1^/gradient = 0.09 kcal mol^−1^ Å ^−1^) Reprinted/adapted with permission from Ref. [89]. 2010, Nunes et al. [89].

**Figure 26 biotech-13-00009-f026:**
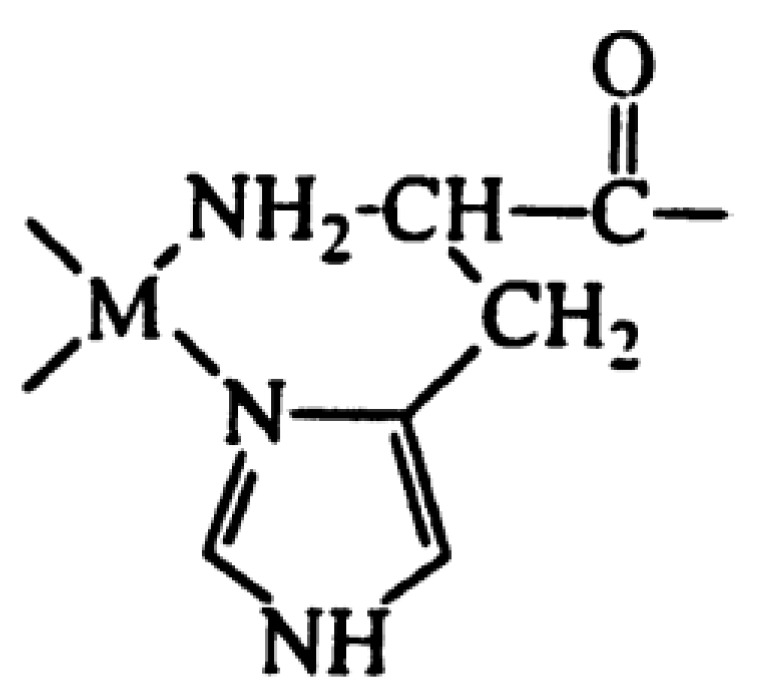
The chelate structure of some metal complexes with Pt(II) and Pd(II) reported by Tsiveriotis et al. Reprinted/adapted with permission from Ref. [94]. 1999, Tsiveriotis et al. [94].

**Figure 27 biotech-13-00009-f027:**
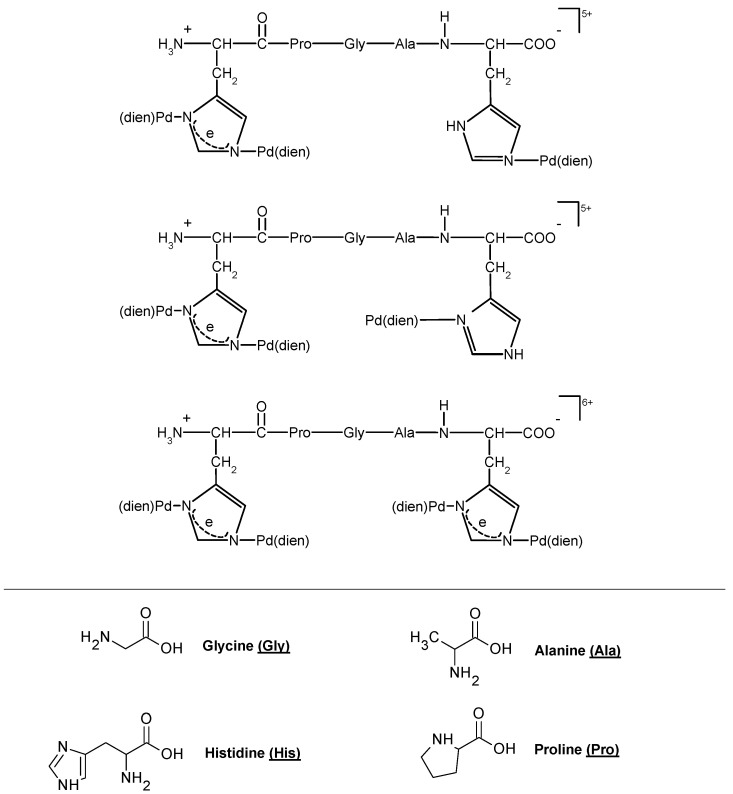
The structure of some metal complexes with Pd(II) Reprinted/adapted with permission from Ref. [94]. 1999, Tsiveriotis et al. [94].

**Figure 28 biotech-13-00009-f028:**
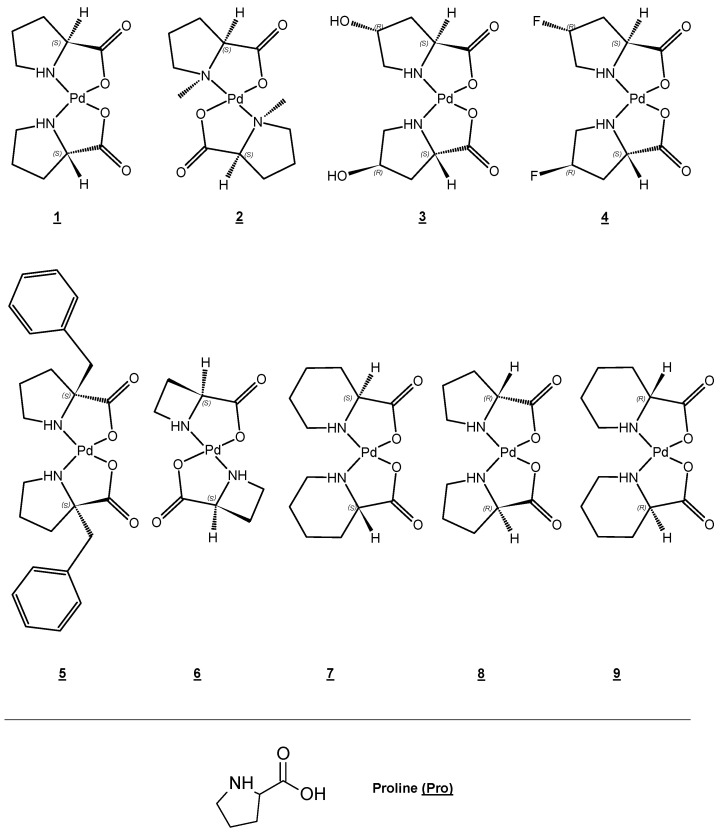
Compound structures and numbering scheme for proline and proline homolog complexes. Stereochemistry is shown at all chiral centers Reprinted/adapted with permission from Ref. [95]. 2019, Hobart et al. [95].

**Figure 29 biotech-13-00009-f029:**
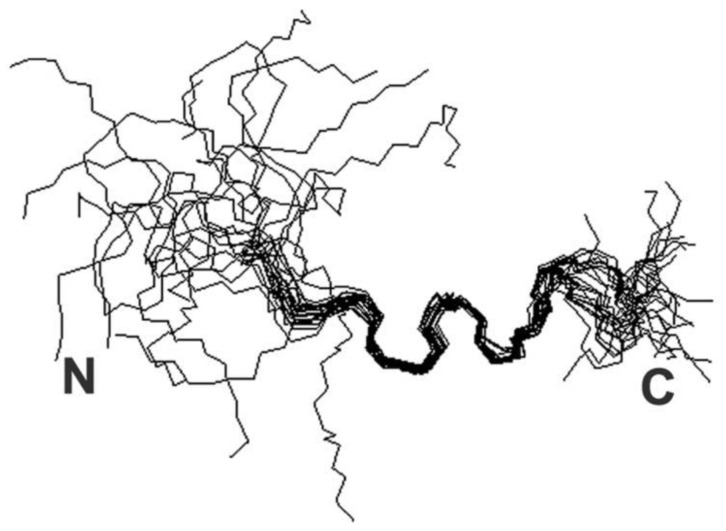
Ensemble of backbone atoms of 20 structures with the lowest target functions for the WT peptide in SDS micellar solution Reprinted/adapted with permission from Ref. [96]. 2013, Wang et al. [96].

**Figure 30 biotech-13-00009-f030:**
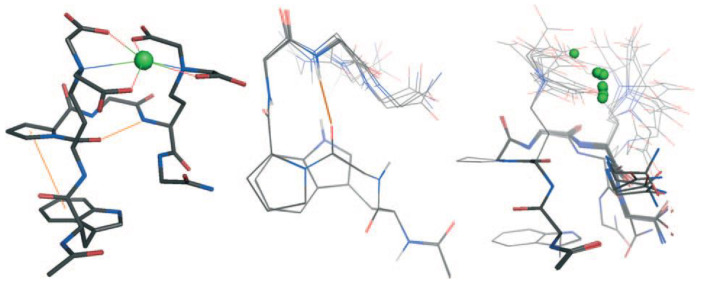
NMR structure of LaP^2^. Left: lowest energy ball-and-stick structure, stabilizing interactions represented as orange lines; middle: superimposition of the backbones and Trp side chains of the 10 lowest energy structures; right: superimposition of the 10 lowest energy structures (backbone and Trp side chain: sticks, coordinating side chain: lines) Reprinted/adapted with permission from Ref. [98]. 2009, Cisnetti et al. [98].

**Table 1 biotech-13-00009-t001:** Various diseases and articles related to them.

Disease	References
Alzheimer’s	[29,40,41,42]
Wilson’s	[43]
Parkinson’s	[44,45]
Menkes	[46]

**Table 2 biotech-13-00009-t002:** Summary table with some essential and non-essential elements.

Essential Elements	References	Non-Essential Elements	References
Cu(I)	[44,47,48,49,50]	Ni(II)	[87,89,90]
Cu(II)	[42,51,52,53,54,55,56]	Pd(II)	[57,92,93,94]
Zn(II)	[31,40,41,69,70,71,72,73,74,75]	Pt(II)	[94]
Co(II) and Co(III)	[32,33,76,77,78,79,80]	Ag(I)	[58,96]
Fe(II) and Fe(III)	[34,35,81,82,83,84,85]	Ga(III)	[78,91]
		La(III), Eu(III), Tb(III)	[98]

**Table 3 biotech-13-00009-t003:** Summary data on the structure of the complexes and the donor atoms involved in the coordination.

Technique	Donor Atom	Metal	Structure	References
1D ^1^H and 2D ^1^H−^1^H TOCSY NMR	S	Cu(I)		[44]
COSY, TOCSY and NOESY, or ROESY	S	Cu(I)		[48]
X-ray crystallography, elemental analysis, UV–Vis, ^1^H-, ^13^C-NMR, LC/MS	N, S	Cu(II)		[42]
1D ^1^H and 2D ^1^H−^1^H TOCSY and NOESY, HMQC, HSQC, ^1^H-^15^N 2D NMR	S	Cu(II)		[51]
UV–Vis, ESI-MS, EPR, COSY, ROESY and TOCSY NMR	O, N	Cu(II)	Square planar or square pyramidal geometries	[52]
UV–Vis, CD, ESR, NMR spectroscopic and MS methods	N	Cu(II)		[54]
Voltametric (cyclic) and spectral (UV–Vis and fluorimetric) analytical techniques, IR, EPR	O, N, S for L1, L3, L6 and O, N for L2, L4	Cu(II)		[56]
UV–Vis, EPR, ^1^H–^1^H TOCSY, ^1^H–^13^C HSQC	N, O	Co(II) and Mn(II)	Octahedral	[79]
^1^H NMR, ^1^H-^1^H TOCSY	N, O	Co(III)		[77]
1D ^1^H and 2D NMR, CD and computational methods	N	Fe(II) and Co(II)		[83]
UV–Vis, 1D ^1^H NMR spectra or 2D soft-COSY experiments	S	Zn(II)		[73]
ESI-MS, TOCSY, NOESY, ROESY, ^1^H-^13^C HSQC and ^1^H-^15^N HSQC NMR	N	Pd(II)	Square planar	[57]
X-ray crystallography, NMR, HRMS	N, O	Pd(II)	Square planar	[95]
ESI–MS, TOCSY and ROESY, ^1^H-^15^N HSQC NMR	N	Al(III)		[86]
UV–Vis, CD, ROESY, TOCSY, NOESY NMR	N	Ni(II) and Cu(II)	Square planar	[87]
^1^H-, ^13^C- and ^195^Pt-NMR	N	Pd(II)	Square planar	[94]

## Data Availability

The data presented in this study are available in this article.

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
