# Peer review of "Synthesis and Biological Activities of Some Metal Complexes of Peptides: A Review"

_biotech, 2024, doi:10.3390/biotech13020009_

Round 1
Reviewer 1 Report
Comments and Suggestions for Authors
Reviewer comments and suggestions
The authors in this study discuss the proteins and peptides that may bound to metals, which is essential to comprehending the biological function of metals. In addition to essential metals, non-essential metals can also interact with the body and have harmful or therapeutic effects. Disorders such as Alzheimer's, Menkes disease, Wilson's disease, and neurodegenerative disorders are caused by abnormalities in metal binding. Additionally, therapeutic peptides—which have antiviral, anticancer, antibacterial, and anti-neurodegenerative properties are becoming more and more popular and discussed in this study
Overall, the manuscript needs to be arranged well. As the authors want to add more or discuss more topics there should be some specific objective that we need to explain well. The citations were not discussed comprehensively. I have a few queries to discuss as well
- Lines 10-12 Please add the method in the abstract part even though it is a review article. Like how they search articles data based and whether if they think it is important one
- Line 18-19 An important conclusion should be added
- Line 31-32 The sentence needs to be modified
- Line 34-35 Both figures are placed or discussed at the same place, for any reason. I think it is better to place an appropriate place and then discuss it.
- Figure 6 The figure was not completely describe
- Line 78 This is no way to cite articles [1,22,23,25,27-28]
- Line 79, at least the authors could discuss a little bit on neurodegenerative about using metal complexes, just citing references did not carry any meaning
- Section 1.2 Please modify the title in a better way so that the reader can easily understand the text without actually going into it
Author Response
Manuscript title: “Synthesis and Biological activities of some metal complexes of peptides: a review”,
authors: Petja Marinova, Kristina Tamahkyarova
Dear Reviewer 1,
Thank you very much for your helpful review. We have accepted all your remarks and the text is properly changed.
Our comments (highlighted in red) are as follows:
Reviewer comments and suggestions
The authors in this study discuss the proteins and peptides that may bound to metals, which is essential to comprehending the biological function of metals. In addition to essential metals, non-essential metals can also interact with the body and have harmful or therapeutic effects. Disorders such as Alzheimer's, Menkes disease, Wilson's disease, and neurodegenerative disorders are caused by abnormalities in metal binding. Additionally, therapeutic peptides—which have antiviral, anticancer, antibacterial, and anti-neurodegenerative properties are becoming more and more popular and discussed in this study
Overall, the manuscript needs to be arranged well. As the authors want to add more or discuss more topics there should be some specific objective that we need to explain well. The citations were not discussed comprehensively. I have a few queries to discuss as well
We added references ref. [1] in abstract part and in all the manuscript [9,10, 37, 38, 39, 40, 66-69, 96, 100-101].
- Lines 10-12 Please add the method in the abstract part even though it is a review article. Like how they search articles data based and whether if they think it is important one
We added this sentence: The research of the better part of the cited papers was conducted using various database plat-forms such as MetalPDB which is available online: https://metalpdb.cerm.unifi.it/getMetalsInPdb [1]. (line 11-13).
- Line 18-19 An important conclusion should be added
We added the important conclusion (line 22-24).
- Line 31-32 The sentence needs to be modified
The sentence is rephrased (line 43-46 in revised manuscript).
- Line 34-35 Both figures are placed or discussed at the same place, for any reason. I think it is better to place an appropriate place and then discuss it.
We deleted this sentence in line 34-35. The figures 1 and 2 have been separated and we added a descriptive paragraph in-between line 37 and 47, respectively.
- Figure 6 The figure was not completely describe
We described completely figure 6 (line 84-93).
- Line 78 This is no way to cite articles [1,22,23,25,27-28]
The sentence is rephrased (line 94-98 in revised manuscript).
- Line 79, at least the authors could discuss a little bit on neurodegenerative about using metal complexes, just citing references did not carry any meaning
We added “The mechanisms of action of these compounds is different and it includes modification of DNA/RNA, permeabilization, protein and cell wall synthesis, and modulation of gradients of cellular membranes.” Line 100-103.
- Section 1.2 Please modify the title in a better way so that the reader can easily understand the text without actually going into it
We modified the title “1.2. Metal complexes of peptides with some essential and non-essential elements”. (line 121)
The corrections in the paper have been made in accordance with the reviewer’s remarks.

Reviewer 2 Report
Comments and Suggestions for Authors
This is a review manuscript about the synthesis and biological activities of some metal complexes of peptides.
The authors should make clear what is the main question addressed in the manuscript.
Also, the authors should underline the gap in the literature that they try to fill.
The authors should present their methodology for selection and inclusion of references. I found that some interesting relevant papers are not cited, but of course might have different priorities. In that case, they should explain their selection procedure.
Can they possibly discuss their differences in approaches from other review that discuss the same general topic?
The visualization in the manuscript is well above average. However, the authors should explain the findings in the protein structures, as not all future readers can be familiar with these drawings.
Also, for all chemical structures shown in figures, please provide the simple chemical formula in the legend of the figure.
After carrying out the above changes, the manuscript can be accepted.
Author Response
Manuscript title: “Synthesis and Biological activities of some metal complexes of peptides: a review”,
authors: Petja Marinova, Kristina Tamahkyarova
Dear Reviewer 2,
Thank you very much for your helpful review. We have accepted all your remarks and the text is properly changed.
Our comments (highlighted in red) are as follows:
This is a review manuscript about the synthesis and biological activities of some metal complexes of peptides.
The authors should make clear what is the main question addressed in the manuscript.
Also, the authors should underline the gap in the literature that they try to fill.
The authors should present their methodology for selection and inclusion of references. I found that some interesting relevant papers are not cited, but of course might have different priorities. In that case, they should explain their selection procedure.
We added references ref. [1] in abstract part and in all the manuscript [9,10, 37, 38, 39, 40, 66-69, 96, 100-101].
Can they possibly discuss their differences in approaches from other review that discuss the same general topic?
This review is hoped to help researchers whose main focus is on the inorganic synthesis and to reveal the hidden potential in ligand molecules like α-amino acids and peptides. It is worth mentioning, that although this review covers literature published in the last two decades, a single review cannot cover all obtained data. While choosing papers we took into consideration the potential opportunities for biological activities that might be applicable to researchers in the field, as well as the nature of the transition metal. In the review we discuss complexes on both traditionally popular metals in essential elements (Cu, Zn, Co, Fe) and non-essential elements (Pd, Pt, Ni, Ag and lanthanides).
It is due to that reason why the data presented in this review is classified by the essential and non-essential elements in these ligands are used, and not by the nature of the ligand molecule. (line 109-120)
The visualization in the manuscript is well above average. However, the authors should explain the findings in the protein structures, as not all future readers can be familiar with these drawings.
We provided additional descriptions to the following figures: 6, 7, 9, 13, 14, 15, 19, 21, 22, 24, 25, 30. We changed figure 12.
Also, for all chemical structures shown in figures, please provide the simple chemical formula in the legend of the figure.
We remade figures 10, 27 and added an additional figure – 28. In all of them we included a legend with the simple chemical formula (below the line). We are unable to provide a simple chemical formula or structure to the rest of the included figures, as peptides are complex molecules.
After carrying out the above changes, the manuscript can be accepted.
The corrections in the paper have been made in accordance with the reviewer’s remarks.

Round 2
Reviewer 1 Report
Comments and Suggestions for Authors
Thanks for the corrections, however, in the abstract part, it was not needed to add citation or references. It should be better to add the point in some related paragraph of the method section rather than adding references in the abstract section. You can add the lines without adding the reference in the abstract. The introduction section should be start with reference number 1. I hope you understand and modify it. Thanks
Author Response
Dear Reviewer 2,
Thank you very much for your helpful review.
We deleted reference 1 in the abstract part.
Reviewer 2 Report
Comments and Suggestions for Authors
The authors have addressed all the issues previously raised and the manuscript can be recommended for acceptance.
Author Response
Dear Reviewer 2,
Thank you very much for your helpful review.
